# The importance of feature preprocessing for differentially private linear optimization

**Ziteng Sun, Ananda Theertha Suresh, Aditya Krishna Menon**
Google Research, New York
{zitengsun,theertha,adityakmenon}@google.com

## Abstract

Training machine learning models with differential privacy (DP) has received increasing interest in recent years. One of the most popular algorithms for training differentially private models is differentially private stochastic gradient descent (DPSGD) and its variants, where at each step gradients are clipped and combined with some noise. Given the increasing usage of DPSGD, we ask the question: is DPSGD alone sufficient to find a good minimizer for every dataset under privacy constraints? Towards answering this question, we show that even for the simple case of linear classification, unlike non-private optimization, (private) feature preprocessing is vital for differentially private optimization. In detail, we first show theoretically that there exists an example where without feature preprocessing, DPSGD incurs an optimality gap proportional to the maximum Euclidean norm of features over all samples. We then propose an algorithm called DPSGD-F, which combines DPSGD with feature preprocessing and prove that for classification tasks, it incurs an optimality gap proportional to the diameter of the features $\max_{x,x' \in D} \|x - x'\|_2$. We finally demonstrate the practicality of our algorithm on image classification benchmarks.

## 1 Introduction

Differential privacy (DP) (Dwork et al., 2014) has emerged as one of the standards of privacy in machine learning and statistics. In machine learning, differentially private methods have been used to train models for language modelling (McMahan et al., 2018), image classification (De et al., 2022), generative diffusion (Ghalebikesabi et al., 2023), and private fine-tuning Yu et al. (2021); Li et al. (2021). We refer readers to Ponomareva et al. (2023) for a detailed survey of techniques and current state of the art methods in private optimization. Following the influential work of Abadi et al. (2016), differentially private stochastic gradient descent (DPSGD) has emerged as one of the most popular algorithms for training private machine learning models and achieves state of the art results in several datasets (De et al., 2022; Ghalebikesabi et al., 2023).

There has also been a recent line of work that focuses on analyzing the theoretical performance of DPSGD and its variants, with a particular focus on convex models (Bassily et al., 2019; Feldman et al., 2020; Bassily et al., 2020; 2021b;a; Song et al., 2021b; Arora et al., 2022). It has been shown that DPSGD and its variants can achieve min-max optimal rates for the task of DP empirical risk minimization and DP stochastic convex optimization under various geometries. Moreover, for convex generalized linear models, DPSGD has also been shown to achieve dimension-independent convergence rate (Song et al., 2021b; Arora et al., 2022). This observation can be extended to general convex models under certain assumptions (Li et al., 2022).

Despite the practical success and appealing theoretical properties of DPSGD, recent empirical results have shown that they may not learn good intermediate features in deep learning image classification tasks (Tramer & Boneh, 2021). In fact, it has been observed in Abadi et al. (2016) that performing a private PCA on the features before performing DPSGD can improve the performance of private training, which highlights that learned features may not be good. This raises a fundamental question:

*Can private feature preprocessing **provably** improve DPSGD?*

There is no clear intuitive answer to this question. On the one hand, feature preprocessing accelerates the convergence of gradient methods in optimization (LeCun et al., 2002), which can help private

optimization since the number of steps is constrained by the privacy requirement. On the other hand, private feature preprocessing will use a portion of the privacy budget, thus decreasing the privacy budget for the DPSGD phase.

As a first step towards answering this question, we show that for the simple task of linear private classification, unlike non-private convex optimization, feature preprocessing is necessary for private optimization to achieve near instance-optimal results. Our findings are as follows.

1. We provide an example where DPSGD with any clipping norm, batch size, and learning rate incurs an error proportional to the maximum Euclidean[1] norm of feature vectors (Section 4).
2. We propose DPSGD-F, a new algorithm that combines both DPSGD and feature preprocessing, and show that the leading term of the error degrades proportional to the diameter of the dataset, which can be significantly smaller than the maximum norm (Section 5). We also complement our result with a near-matching information-theoretic lower bound (Section 6).
3. We empirically validate our findings on a few standard datasets and show that DPSGD-F outperforms DPSGD on a subset of the datasets. For the task of privately finetuning the last layer of a pretrained model (using ImageNet1K) on CIFAR-100 under $\varepsilon = 1$, our result improves the previous accuracy from 70.6% (De et al., 2022) to 71.6% (Section 7).

The rest of the paper is organized as follows. In Section 2, we outline the problem setting and in Section 3 discuss prior work and our contribution. In Section 4 we provide a counter-example for DPSGD and in Section 5, we describe our new algorithm and its performance. In Section 6, we provide an information theoretic lower bound. Finally, in Section 7, we demonstrate the practicality of the proposed algorithm on image clssification tasks.

## 2 PRELIMINARIES

**Classification.** Let $\mathcal{X}$ denote the feature space and $\mathcal{Y} = \{-1, 1\}$ be the set of labels. Unless otherwise specified, we set $\mathcal{X}$ to be a ball of radius $R$ in $d$-dimensional space denoted by $\mathcal{B}_2^d(R)$. Let $h$ be a classifier that takes parameter $\theta$ and feature $x$ and computes a score $h(\theta, x) \in \mathbb{R}$. The performance of the classifier is measured by a loss function $\ell \colon \mathbb{R} \times \mathcal{Y} \to \mathbb{R}_+$. We assume $\ell$ is a *margin loss* (Bartlett et al., 2006) of the form

$$\ell(h(\theta, x), y) = \phi(y \cdot h(\theta, x)), \tag{1}$$

where $\phi \colon \mathbb{R} \to \mathbb{R}$ is typically a convex, non-increasing function with a bounded value $\phi(0) < \infty$ at 0. Canonical examples of $\phi$ include the *hinge loss* $\phi(z) = [1 - z]_+$, and the *logistic loss* $\phi(z) = \log(1 + e^{-z})$.

Let $D = \{(x_i, y_i)\}_{i \in [n]}$ be a training set of size $n$. Typically, the goal is to learn a classifier by minimizing the average loss over all samples given by

$$L(\theta, D) \triangleq \frac{1}{n} \sum_{i \in [n]} \ell(h(\theta, x_i), y_i).$$

We are interested in linear classification models that are described as follows. Let $\theta = (w, b) \in \mathbb{R}^{d+1}$, where $w \in \mathbb{R}^d$ and $b \in \mathbb{R}$, and $h((w, b), x_i) = w \cdot x_i + b$. We further assume that $\ell(h(\theta, x), y)$ is convex in $\theta$, and that $\forall y, \ell(\cdot, y)$ is convex and $G$-Lipschitz in the first argument. The model can be viewed as a generalized linear model (GLM), with additional restrictions on the loss function. We denote the minimizer of the empirical loss by

$$\theta_D^* \in \arg\min L(\theta, D).$$

We will drop the subscript when the dataset is clear from context. The following assumption on the minimizer will be useful in our performance analysis. It assumes that the minimizer is not a trivial solution which does not separate any two data points in the dataset. This is a reasonable assumption as long as $\ell$ is a good proxy loss for classification.

**Assumption 1** (Nontrivial minimizer). The dataset $D$ and loss function $\ell$ satisfies that at the minimizer $\theta^*$, there exists $x, x' \in D$ such that

$$h(\theta^*, x) \cdot h(\theta^*, x') \le 0.$$

---

[1]When unspecified, we use norm to refer to the Euclidean norm by default.

---

**Algorithm 1** Differentially private SGD (Abadi et al., 2016)

---

**Input:** Input: Dataset $D = \{(x_1, y_1), (x_2, y_2), \ldots, (x_n, y_n)\}$ of $n$ points, privacy parameter $\varepsilon, \delta$, clipping norm $C$, step size $\eta$, number of iterations $T$, batch size $B \geq \max\{n\sqrt{\varepsilon/4T}, 1\}$,

1: Set $\sigma^2 = \frac{8TC^2 \log(1/\delta)}{n^2 \varepsilon^2}$.
2: Choose an inital point $\theta_0$.
3: **for** $t = 0, 1, \ldots, T - 1$ **do**
4:     Sample a batch $B_t$ of $B$ data points with replacement.
5:     For all $i \in B_t$, compute the gradient $\nabla \ell(h(\theta, x_i), y_i)$.
6:     Compute a noisy clipped mean of the gradients by

$$\hat{g}_t = \frac{1}{|B_t|} \sum_{i \in B_t} \left( \text{Clip}(\nabla \ell(h(\theta_t, x_i), y_i)), C) + \mathcal{N}(0, \sigma^2 \mathbb{I}_d) \right) \tag{2}$$

7:     Update the parameter by $w_{t+1} = w_t - \eta \hat{g}_t$.
8: **end for**
9: **Return** $\bar{T} = \frac{1}{T} \sum_{t=1}^{T} w_t$.

---

**Differential privacy (DP) (Dwork et al., 2014).** DP requires the optimization algorithm to output similar outputs for similar training datasets. More precisely, differential privacy is defined below.

**Definition 1** (Differential privacy). Let $\mathcal{D}$ be a collection of datasets. An algorithm $\mathcal{A} : \mathcal{D} \to \mathcal{R}$ is $(\varepsilon, \delta)$-DP if for any datasets $D$ and $D'$ that differ by only one data point, denoted as $|D \Delta D'| = 1$, and any potential outcome $O \in \mathcal{R}$, the algorithm satisfies

$$\Pr\left(\mathcal{A}(D) \in O\right) \leq e^\varepsilon \Pr\left(\mathcal{A}(D') \in O\right) + \delta.$$

Our goal is to find a $\theta_{\text{prv}}$ that is $(\epsilon, \delta)$-differentially private and minimizes the optimality gap w.r.t. the empirical risk, which is also referred to as the privacy error,

$$\mathbb{E}[L(\theta_{\text{prv}}, D)] - \min_\theta L(\theta, D),$$

where the expectation is over the randomness of the private algorithm.

**Notations.** We use $\text{Clip}(x, C)$ to denote the clipping operation with norm $C$, defined by

$$\text{Clip}(x, C) := \min \left\{ 1, \frac{C}{\|x\|_2} \right\} \cdot x.$$

For a vector $x$ of dimension $d$, we use $(x, 1)$ to refer to the $(d+1)$-dimensional vector obtained from augmenting 1 to $x$. We use $\text{diam}(D) := \max_{x, x' \in D} \|x - x'\|_2$ to denote the diameter of features in the dataset $D$. Given a set of features $x_1, x_2, \ldots, x_n$ from a dataset $D$, let $U(D)$ denote the eigenbasis of $\sum_i x_i x_i^\top$. Let $M(D)$ be the projection operator to this eigenspace $U(D)$, given by $M(D) = U(D)(U(D))^\top$. Then, $M(D)$ defines a seminorm[2] given by

$$\|v\|_{M(D)} = \|v M(D) v^\top\|_2.$$

## 3    RELATED WORK AND OUR CONTRIBUTION

**Differentially private stochastic gradient descent (DPSGD).** We start by describing the mini-batch variant of the DPSGD algorithm in Algorithm 1. The DPSGD algorithm is a modification of the popular SGD algorithm for training learning models. In each round, the individual sample gradients are clipped and noised to bound the per-round privacy loss (the influence of one data point). The overall privacy guarantee combines tools from privacy amplification via subsampling and strong composition (see Abadi et al. (2016)).

DPSGD has shown to achieve dimension-independent convergence result for optimizing generalized linear models. The upper bound below is implied by Song et al. (2020, Theorem 4.1).

---

[2]Given a vector space $V$ over real numbers, a seminorm on $V$ is a nonnegative-valued function $Q$ such that for all $x \in \mathbb{R}$ and $v \in V$, $Q(x \cdot v) = |x| \cdot Q(v)$ and for all $u, v \in V$, $Q(u + v) \leq Q(u) + Q(v)$.

**Lemma 1** (Song et al. (2020))**.** *There exists an* $(\varepsilon, \delta)$*-DP instance of DPSGD, whose output satisfies,*

$$\mathbb{E}[L(\theta_{\mathrm{prv}}, D)] - L(\theta_D^*, D) \leq 2G\|\theta^*\|_M R\left(\frac{\sqrt{\mathrm{rank}(M)\log(1/\delta)}}{n\varepsilon}\right),$$

*where $G$ is the Lipschitz constant of the loss function, $R = \max_i \|x_i\|_2$ and $M = M(D')$ with $D' = \{(x, 1) \mid x \in D\}$.*

Note that in the analysis in Song et al. (2020), the norm of the gradients is bounded by $G\sqrt{R^2 + 1}$, and the clip norm is chosen such that the gradients are not clipped. The above result is shown to be minimax optimal in terms of the dependence on the stated parameters. Recall that $\mathrm{diam}(D) := \max_{x,x' \in D} \|x - x'\|_2$ denotes the diameter of feautures in the dataset $D$. So a natural question is to ask is whether the dependence on $R$ can be improved to $\mathrm{diam}(D)$, which can be useful in cases when dataset if more favorable *e.g.,* when $\mathrm{diam}(D) \ll R$. If yes, can the improvement be achieved by DPSGD?

We answer the first question by proposing an algorithm that combines feature preprocessing and DPSGD, and improves the dependence on $R$ to $\mathrm{diam}(D)$ in the leading term of the optimality gap, stated below.

**Theorem 1.** *There exists an $(\varepsilon, \delta)$-differentially private algorithm DPSGD-F, which is a combination of private feature preprocessing and DPSGD, such that when $n = \Omega\left(\frac{\sqrt{d\log(1/\delta)}\log R\log(d)}{\varepsilon}\right)$ and $\varepsilon = O(1)$, the output $\theta_{prv}$ satisfies*

$$\mathbb{E}[L(\theta_{prv}, D)] - L(\theta^*, D)$$
$$= O\left(G\|\theta^*\|_M\left(\mathrm{diam}(D) + \frac{R}{n^2}\right)\left(\frac{\sqrt{\mathrm{rank}(M)\log(1/\delta)} + \log n}{n\varepsilon}\right) + \frac{\phi(0)\log(n)}{n\varepsilon}\right),$$

*where $M = M(D')$ with $D' = \{(x, 1) \mid x \in D\}$. As discussed before Lemma 3, $\frac{R}{n^2}$ can be reduced to any inverse polynomial function of $n$ by increasing the requirement on $n$ by a constant factor.*

We will describe the algorithm and disucss the proof in Section 5. The next theorem states that the first term in the above result is tight.

**Theorem 2.** *Let $\mathcal{A}$ be any $(\varepsilon, \delta)$-DP optimization algorithm with $\varepsilon \leq c$ and $\delta \leq c\varepsilon/n$ for some constant $c > 0$. There exists a dataset $D = \{(x_i, y_i), i \in [n]\}$ and a loss function $\ell(\theta \cdot x, y)$ that is convex and $G$-Lipschitz loss functions for all $y$, which is of form Eq. (1), and*

$$\mathbb{E}\left[L(\mathcal{A}(D), D)\right] - L(\theta_D^*, D) = \Omega\left(G \cdot \mathrm{diam}(D) \cdot \min\left\{1, \frac{\|\theta^*\|_M\sqrt{\mathrm{rank}(M)}}{n\varepsilon}\right\}\right),$$

*where $M = M(D')$ with $D' = \{(x, 1) \mid x \in D\}$.*

Moreover, we show that DPSGD must incur an error proportional to $R$ by providing a counter-example in Section 4.

## 4 A COUNTER-EXAMPLE FOR DPSGD

We consider the simple case of binary classification with hinge loss. Let $\mu > 0$. Let $e_1$ and $e_2$ denote two orthogonal basis vectors. Suppose dataset $D'$ is partitioned into two equal parts $D'_1$ and $D'_{-1}$, each with size $n/2$. $D'_1$ contains $n/2$ samples with $x = \mu e_1 + e_2$ and $y = 1$ and $D'_{-1}$ contains $n/2$ samples with $x = \mu e_1 - e_2$ and $y = -1$. A visualization of the dataset is provided in Fig. 1. Similarly, let the dataset $D''$ is partitioned into two equal parts $D''_1$ and $D''_{-1}$, each with size $n/2$. $D''_1$ contains $n/2$ samples with $x = \mu e_1 + e_2$ and $y = -1$ and $D''_{-1}$ contains $n/2$ samples with $x = \mu e_1 - e_2$ and $y = 1$. We assume there is no bias term for simplicity, *i.e.,* $b = 0$[3]. For any $w = (w(1), w(2))$, let

$$\ell(w \cdot x, y) = \max\{1 - y(w \cdot x), 0\}.$$

---

[3]The bias term can be added by assuming there is an additional entry with a 1 in the feature vector. The proof will go through similarly as the optimal parameter has $b = 0$.

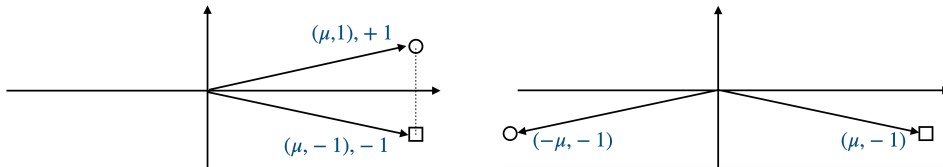

Figure 1: Feature vectors.                    Figure 2: Gradient vectors.

Further, the average empirical loss on $D'$ will be

$$L(w, D') = \frac{1}{2}\max\{1 - (\mu w(1) + w(2)), 0\} + \frac{1}{2}\max\{1 + (\mu w(1) - w(2)), 0\}.$$

Observe that $w^{*'} = (0, 1)$ has zero empirical loss for for dataset $D'$ and $w^{*''} = (0, -1)$ has zero empirical loss for for dataset $D''$. The next theorem states that DPSGD with any clipping norm, batch size, number of steps, learning rate and any initialization will not obtain nontrivial error on both $D'$ and $D''$ when $\mu > n\varepsilon$.

**Theorem 3.** *Let $\mathcal{A}$ be a $(\varepsilon, \delta)$-private DPSGD algorithm with any number of steps $T$, learning rate $\eta$, clipping norm $C$, and (possibly randomized) initialization $w_0$. We have when $n < \mu/\varepsilon$,*

$$\max_{D \in \{D', D''\}} \Big( \mathbb{E}\left[L(\mathcal{A}(D), D)\right] - \min_w L(w, D) \Big) = \Omega(1).$$

The crux of the proof involves showing that if the initialization is "uninformative" for a dataset, then DPSGD does not obtain nontrivial error when $\mu > n\varepsilon$. Here by "uninformative", we mean $w_0 = (w_0(1), w_0(2))$ satisfy that $w_0(2) \leq 0$ for dataset $D'$ and $w_0(2) \geq 0$ for dataset $D''$. Note that at least one of the above two conditions must hold. Below we give a high-level idea of the proof. Since $D'_1$ and $D'_{-1}$ (and similarly $D''_1$ and $D''_{-1}$) have the same $x(1)$, the performance of the final optimizer will depend mostly on the second parameter $w(2)$. The following lemma can be proved.

**Lemma 2.** *For dataset $D'$, let $w$ be any iterate of the DPSGD algorithm, then*

$$\mathbb{E}\left[L(w, D')\right] - \min_w L(w, D') \geq \Pr\left(w(2) \leq 0\right).$$

*Similarly, for dataset $D''$,*

$$\mathbb{E}\left[L(w, D'')\right] - \min_w L(w, D'') \geq \Pr\left(w(2) \geq 0\right).$$

We will focus on the case when $w_0(2) \leq 0$ and consider dataset $D'$ and loss $L(w, D')$[4]. It would be enough to show that for every iteration $t$, the parameter $w_t$ has a constant probability of being negative. The gradient of each individual loss functions satisfies that for $(x, y) \in D'$, we have

$$\nabla_w \ell(w \cdot x, y) = \begin{cases} \mathbb{1}\{\mu w(1) + w(2) < 1\} \cdot (-\mu, -1) & \text{if } (x, y) \in D'_1, \\ \mathbb{1}\{-\mu w(1) + w(2) < 1\} \cdot (\mu, -1) & \text{if } (x, y) \in D'_{-1}. \end{cases}$$

Hence the norm of the gradient vectors can be as large as $\sqrt{\mu^2 + 1}$. By clipping the gradient to norm $C$, the 'signal' component of the gradient on the second coordinate will be decreased to $\min\{1, C/\sqrt{\mu^2 + 1}\}$. However, in each iteration, DPSGD will add a Gaussian noise with standard deviation proportional to $C/n\varepsilon$, which can be large compared to $\min\{1, C/\sqrt{\mu^2 + 1}\}$ when $\mu$ is large. When $n < \mu/\varepsilon$, the total noise will dominate the signal, making the probability of $w_t(2) < 0$ nontrivial. We provide the complete proof in Appendix A.

**DPSGD with better mean estimation algorithm.** One attempt to solve the above issue is to replace Eq. (2) by recently proposed private mean estimation algorithms, which can improve the estimation error to scale with the diameter of the gradients (*e.g.,* in Karwa & Vadhan (2017); Huang et al. (2021)). However, it is unclear whether this will work. As shown in Fig. 2, the diameter of the gradients can be proportional to the maximum norm of feature vectors instead of the diameter.

---

[4]When $w_0(2) \geq 0$, we will consider $D''$ and loss $L(w, D'')$ instead.

---

**Algorithm 2** DPSGD with feature preprocessing (DPSGD-F).

---

**Input:** Input: Dataset $D = \{(x_1, y_1), (x_2, y_2), \ldots, (x_n, y_n)\}$ of $n$ points, $\varepsilon, \delta$. Private mean estimation algorithm $\mathrm{PrivMean}$ (Lemma 3). Private quantile esimation algorithm $\mathrm{PrivQuantile}$ (Lemma 4), DPSGD for GLM (Lemma 1).

1: **Private mean estimation.** Compute $\hat{\mu}$, the differentailly private mean of features in $D$ with Lemma 3 and privacy budget $(\varepsilon/3, \delta/3)$.

2: **Private quantile estimation.** Let $S = \{\|x - \hat{\mu}\|_2, x \in D\}$ be the set of distances from the feature vectors to the computed mean. Find a private quantile $\tau$ of set $S$ using Lemma 4 with privacy budget $(\varepsilon/3, \delta/3)$.

3: **Translate and augment the dataset with bias.** Preprocess the dataset by translation and appending a constant feature $\tau$. Let $D' = \{(x'_1, y_1), \ldots, (x'_n, y_n)\}$ where

$$x'_i = (x_i - \hat{\mu}, \tau).$$

4: **Feature clipping.** Let $D'' = \{(x''_1, y_1), (x''_2, y_2), \ldots, (x''_n, y_n)\}$, where

$$x''_i = \mathrm{Clip}\left(x'_i, \sqrt{2}\tau\right).$$

5: **DPSGD with preprocessed features.** Compute an approximate minimizer $\theta''_{\mathrm{prv}} \in \mathbb{R}^{d+1}$ of $L(\theta, D'')$ using the DPSGD algorithm for GLM in Lemma 1 with $(\varepsilon/3, \delta/3)$.

6: **Return** $\theta_{\mathrm{prv}} = (w_{\mathrm{prv}}, b_{\mathrm{prv}})$, where $w_{\mathrm{prv}} = \theta''_{\mathrm{prv}}[1 : d]$ and $b_{\mathrm{prv}} = \theta''_{\mathrm{prv}}[d + 1]\tau - \hat{\mu}$.

---

In the next section, we take advantage of the fact that the data points are close in the feature space, and design an algorithm that first privately preprocesses the features and then perform DPSGD to achieve better bounds. In particular, the bound in Theorem 1 states that when $\max\{\sqrt{\log(\mu)\log(1/\delta)}/\varepsilon, \sqrt[3]{\mu/\varepsilon}\} \ll n < \mu/\varepsilon$, the new algorithm achieves a $o(1)$ error, which is strictly better than DPSGD.

## 5 A FEATURE PREPROCESSING AUGMENTED DPSGD ALGORITHM

We propose a new algorithm called DPSGD-F (Algorithm 2) that combines feature preprocessing and DPSGD. We show that the new algorithm can achieve the performance described in Theorem 1. We overview each step of the algorithm below and provide theoretical guarantees for each step. The detailed proofs are listed in Appendix B.

**Private mean estimation of features.** We first compute a differentially private mean of features $\hat{\mu}$ using the instance-optimal mean estimation algorithm from Huang et al. (2021), who showed that the mean can be estimated efficiently with error proportional to the diameter of the dataset. The following result on private mean estimation follows by combining Theorem 3.3 and Remark 3.2 from Huang et al. (2021). Note that the result of Huang et al. (2021) is a high-probability bound, which does not explicitly depend on $R$ (or $r(D)$ in their notation). To obtain an in-expectation bound, we choose a failure probability of $1/n^2$ in the proof and this results in the $R/n^2$ term in Lemma 3. This term can be reduced to any inverse polynomial function of $n$ by increasing the requirement on $n$ in the assumption by a constant factor. We choose $1/n^2$ here mainly for presentation purpose. See Appendix B.1 for the detailed analysis.

**Lemma 3** (Appendix B.1). *Let $\varepsilon \leq \log(3/\delta)$ and $n \geq c \cdot \left( \frac{\sqrt{d\log(1/\delta)\log R}\log(d)}{\varepsilon} \right)$ for a sufficiently large constant c, then there exists an $(\varepsilon/3, \delta/3)$-DP algorithm whose output $\hat{\mu} \in \mathcal{B}_2^d(R)$ satisfies*

$$\mathbb{E}[\|\mu - \hat{\mu}\|_2] \leq \mathrm{diam}(D) + \frac{2R}{n^2}.$$

The above lemma implies that if $n$ is large enough, $\hat{\mu}$ is a good approximation for $\mu$ and hence

$$\forall x_i \in D, \qquad \mathbb{E}[\|x_i - \hat{\mu}\|_2] \leq 2\mathrm{diam}(D) + \frac{2R}{n^2}. \tag{3}$$

Existing algorithms for differentially private ERM require the knowledge of $\mathrm{diam}(D)$, which is unknown. Hence, we compute an estimate of $\mathrm{diam}(D)$ by private quantile estimation.

**Private quantile estimation.** Changing one sample in a dataset can change the diameter by $R$. Hence, the sensitivity of $\mathrm{diam}(D)$ is $R$ and it is difficult to estimate it privately with good accuracy. Instead we estimate the $\tilde{O}(1/(n\varepsilon))$ quantile of $\|x_i - \hat{\mu}\|_2$ using Dick et al. (2023, Theorem 2).

**Lemma 4** (Appendix B.2). *There exists a $(\varepsilon/3, \delta/3)$ algorithm that finds a threshold $\tau$ such that*

$$\mathbb{E}[\tau|\hat{\mu}] \le 2 \max_i \|x_i - \hat{\mu}\|_2 + \frac{2R}{n^2}.$$

*Let $\varepsilon = O(1)$ and $Z_\tau$ be the number of points such that $\|x_i - \hat{\mu}\|$ is larger than $\tau$, then $\mathbb{E}[Z_\tau \mid \hat{\mu}] \le \frac{125}{\varepsilon} \log n$.*

**Translated and augment dataset with bias.** Another issue with the above mentioned approach is the scale of the bias term. If $\theta = (w, b)$, then the gradient with respect to $w$ scales linearly in $x$, but is a fixed constant for $b$. This causes issues in the theoretical analysis. Hence, we construct an augmented dataset $D'$ where $x_i' = (x_i - \hat{\mu}, \tau)$. Note that $x_i' \in \mathbb{R}^{d+1}$. We also remove the bias term when optimizing over $D'$ since we have already included a bias term in the augmented features. This also guarantees that the norm of the gradient is bounded by $\Theta(\tau)$ with high probability. We show that we are not losing any information in translation and augmentation and hence the minimum value of empirical risk should remain the same. We formalize it in the next lemma.

**Lemma 5.** *For any $\theta = (w, b)$, let $\theta' = (w, (b + w \cdot \hat{\mu})/\tau)$, we have*

$$L(\theta, D) = L(\theta', D').$$

*In particular, $(w^*, b^*)$ is a minimizer of $D$ if and only if $(w^*, (b^* + \hat{\mu} \cdot w^*)/\tau)$ is a minimizer of $D'$.*

*Proof.* To prove the first result, observe that for any point $(x, y) \in D$,

$$
\begin{aligned}
\ell(h((w, b), x), y)) &= \phi(y \cdot (w \cdot x + b)) \\
&= \phi(y \cdot (w \cdot (x - \hat{\mu}) + b + w \cdot \hat{\mu})) \\
&= \phi(y' \cdot (w \cdot x' + (b + w \cdot \hat{\mu})/\tau \cdot \tau)) \\
&= \ell(h((w, (b + w \cdot \hat{\mu})/\tau), x'), y').
\end{aligned}
$$

The second result follows by taking the minima over all $(w, b)$ in the first result. $\qquad\square$

Recall that results on privacy error involve the projector on the design matrix and the norm of the minimizer. Hence, we next provide connections between the rank of $M(D)$ and $M(D')$ and the norms of the minimizers of $D$ and $D'$. The proof mainly follows from the fact that the translating and augmenting operations are performing the same linear operation on each data point, which won't change the minimizer up to scaling and the bias term.

**Lemma 6** ((Appendix B.3)). *Let $M(D)$ and $M(D')$ be the projectors design matrices of non-augmented dataset $D$ and augmented dataset $D'$ respectively, then*

$$\mathrm{rank}(M(D')) \le \mathrm{rank}(M(D)) + 5.$$

*Furthermore, let $\theta^*$ and $\theta'^*$ be the empirical risk minimizers of $D$ and $D'$ respectively. If Assumption 1 holds then*

$$\mathbb{E}[\|\theta'^*\|_2 \cdot \tau|\hat{\mu}] \le 2\|\theta^*\|_2 \max_{x_i \in D} \|\hat{\mu} - x_i\|_2 + \frac{2\|\theta^*\|_2 R}{n^2}.$$

**Feature clipping.** As mentioned above, due to errors in previous steps, the $\ell_2$ norm of the features may not be bounded with a small probability. Hence, we clip the augmented features from dataset $D'$ to obtain a dataset $D''$. Similar to the Lemma 6, we relate the rank of $M(D')$ and $M(D'')$ next. The proof mainly follows from the fact that clipping will only change the scale of each feature vector but not their direction.

**Lemma 7** (Appendix B.4). *Let $M(D')$ and $M(D'')$ be the projectors to the design matrices of datasets $D'$ and $D''$ respectively, then*

$$\mathrm{rank}(M(D'')) = \mathrm{rank}(M(D')).$$

**DPSGD with preprocessed features.** We next solve ERM using DPSGD algorithm on $D''$. Since norm of the features are bounded by $\sqrt{2}\tau$, the gradients will be bounded by $\sqrt{2}G\tau$. Furthermore, since the dataset is augmented, we don't include the bias term and treat weights as a vector in $\mathbb{R}^{d+1}$. The guarantee of DPSGD on these features follows from previous work Song et al. (2020, Theorem 4.1). Similar to Song et al. (2020), the clipping norm is chosen to be $\sqrt{2}G\tau$ so that no gradients are clipped in all steps.

**Reverting the solution to the original space**. The output of the DPSGD algorithm is in $\mathbb{R}^{d+1}$, and furthermore only works on the translated and augmented dataset $D''$. Hence, we translate it back to the original space, by appropriately obtaining $w_{\text{prv}}$ and rescaling the bias.

The computation complexity of the algorithm is the same as DPSGD since the preprocessing algorithms can all be implemented in time $\tilde{O}(nd)$, which is less than that of the DPSGD phase. The proof of privacy guarantee in Theorem 1 follows directly from the composition theorem (Dwork et al., 2014, Theorem 3.16). The utility guarantee follows from the above stated results and we provide the complete proof of in Appendix B.5.

## 6   LOWER BOUND

In this section, we prove Theorem 2, which shows that the performance of our algorithm is tight up to logarithmic factors. The proof follows almost immediately from Theorem 3.3 in Song et al. (2021a), with a slight modification to the label and loss function to make sure that it is a valid classification problem of form Eq. (1). We first state the result in Song et al. (2021a) (a restated version of Theorem 3.3) below.

**Theorem 4** (Song et al. (2021a)). *Let $\mathcal{A}$ be any $(\varepsilon, \delta)$-DP algorithm with $\varepsilon \leq c$ and $\delta \leq c\varepsilon/n$ for some constant $c > 0$. Let $1 \leq d_0 \leq d$ be an integer. Let $\ell$ be defined by $\ell(\theta \cdot x, y) = |\theta \cdot x - y|$. Then there exists a data set $D = \{(x_i, y_i), i \in [n]\}$ such that the following holds. For all $i \in [n]$, $x_i \in \{(0, \ldots, 0), (1, 0, \ldots, 0), (0, 1, 0, \ldots, 0), (0, \ldots, 0, 1)\}$ and $y_i \in \{0, 1\}$. Let $\theta^* = \underset{\theta \in \mathbb{R}^d}{\arg\min} L(\theta; D)$ (breaking ties towards lower $\|\theta\|_2$). We have $\|\theta^*\|_2 \in [0, 1]^d$, and*

$$\mathbb{E}\left[L\left(\text{Proj}_{[0,1]^d}(\mathcal{A}(D)); D\right)\right] - L(\theta^*; D) \geq \Omega\left(\min\left\{1, \frac{d_0}{\varepsilon n}\right\}\right),$$

*where $\text{Proj}_{[0,1]^d}(\mathcal{A}(D))$ denotes the projection of $\mathcal{A}(D)$ to $[0, 1]^d$.*[5]

**Proof of Theorem 2:**   Take the dataset $D = \{(x_i, y_i), i \in [n]\}$ from Theorem 4, we can construct a dataset $D'$ as following. $\forall i$, let $x_i' = x_i, y_i' = 2y_i - 1$. Set $\ell'(\theta \cdot x, y) = \max\{1 - y(2\theta \cdot x - 1), 0\}$, which is of the form Eq. (1). Then, for all $\theta \in [0, 1]^d$, we have $\theta \cdot x \in [0, 1]$ and

$$\ell'(\theta \cdot x, y) = \begin{cases} 2 - 2\theta \cdot x = |2\theta \cdot x - 1 - y| & \text{if } y = +1, \\ 2\theta \cdot x = |2\theta \cdot x - 1 - y| & \text{if } y = -1. \end{cases}$$

Hence $\forall i \in [n], \ell'(\theta \cdot x_i', y_i') = |2\theta \cdot x_i' - 1 - y_i'| = 2|\theta \cdot x_i - y_i|$. Moreover, it can be seen that $\forall x \in [0, 1]^d, y \in \{+1, -1\}$

$$\ell'(\text{Proj}_{[0,1]^d}(\theta) \cdot x, y) \leq \ell'(\theta \cdot x, y). \tag{4}$$

Hence the minimizer of $L'(\theta; D')$ lies in $[0, 1]^d$. The above facts imply that $\theta^*$ is also the minimizer of $L'(\theta; D')$, and by Theorem 4,

$$\mathbb{E}\left[L'\left(\text{Proj}_{[0,1]^d}(\mathcal{A}(D')), D'\right)\right] - L'(\theta^*; D') \geq \Omega\left(\min\left\{1, \frac{d_0}{\varepsilon n}\right\}\right).$$

Together with Eq. (4), we have

$$\mathbb{E}[L'(\mathcal{A}(D'), D')] - L'(\theta^*; D') \geq \Omega\left(\min\left\{1, \frac{d_0}{\varepsilon n}\right\}\right).$$

---

[5]Theorem 3.3 in Song et al. (2021a) is stated without the projection operator while in fact the projection operator is used throughout the proof of Theorem B.1 and Theorem 3.3. Hence the stated result holds with no change in the proof.

Note that for all $\theta \in \mathbb{R}^d, \forall i \in [n], \|\partial_\theta \ell'(h(\theta, x_i'), y_i')\|_2 \leq 2\|x_i\|_2 \leq 2$. And the dataset satisfies that $\text{diam}(D) \leq \sqrt{2}, \text{rank}(\sum_{i=1}^n x_i x_i^\top) \leq d_0$. Moreover, $\|\theta^*\|_M \leq \|\theta^*\|_2 \leq \sqrt{d_0}$. Hence we have

$$\mathbb{E}\left[L'\left(\mathcal{A}(D'), D'\right)\right] - L'\left(\theta^*; D'\right) \geq \Omega\left(\text{diam}(D') \min\left\{1, \frac{\|\theta^*\|_M \sqrt{\text{rank}(M)}}{n\varepsilon}\right\}\right).$$

The proof can be generalized to a general Lipschitz constant $G$ and $\text{diam}(D)$ by rescaling. $\quad\square$

## 7 Experiments

We empirically demonstrate our findings by evaluating DPSGD and our proposed feature-normalized DPSGD (DPSGD-F[6]) for the task of training a linear classifier on three popular image classification datasets: (1) MNIST (Lecun et al., 1998); (2) Fashion-MNIST (Xiao et al., 2017); (3) CIFAR-100 (Krizhevsky et al., 2009) with pretrained features. For MNIST and Fashion-MNIST, we directly train a linear classifier with the pixel-format images as inputs. For CIFAR-100, we use the features obtained from the representations in the penultimate layer in a WRN model pretrained on ImageNet (De et al., 2022). In this case, training a linear classifier using these features is equivalent to fine-tuning the last layer of the WRN model.

Table 1: Accuracy comparison on different datasets with different values of $\varepsilon$ and $\delta = 10^{-5}$. † CIFAR-100 is trained and tested using features pretrained on ImageNet (De et al., 2022). Each number is an average over 10 independent runs. The number in the parenthesis represents the standard deviation of the accuracy under the optimal parameter settings.

| Dataset | Accuracy ($\varepsilon = 1$) | | Accuracy ($\varepsilon = 2$) | | Non-private acc. |
|---|---|---|---|---|---|
| | DPSGD | DPSGD-F | DPSGD | DPSGD-F | SGD |
| MNIST | 87.4 (0.1) | **92.0** (0.1) | 89.6 (1.2) | **92.3** (0.1) | 92.9 (0.1) |
| FMNIST | 77.2 (0.4) | **84.0** (0.1) | 78.7 (0.4) | **84.5** (0.2) | 85.0 (0.1) |
| CIFAR-100 (pretrained)† | 70.6 (0.3) [7] | **71.6** (0.2) | 73.8 (0.2) | **74.4** (0.2) | 78.5 (0.1) |

Similar to De et al. (2022), for each combination of dataset, privacy parameter, and algorithm, we report the best accuracy obtained from a grid search over batch size, steps size, and number of epochs[8]. For DPSGD-F, we also did a grid search over the allocation of privacy budget between the preprocessing phase and the DPSGD phase. We leave choosing the parameters automatically as a future direction. Detailed implementation and parameter settings are listed in Appendix C.

Our results are listed in Table 1. Our proposed algorithm consistently improves upon DPSGD for all three datasets under $\varepsilon = 1$ and $\varepsilon = 2$, which empirically demonstrates our theoretical findings.

## 8 Discussion

In this work, we ask the fundamental question of whether DPSGD alone is sufficient for obtaining a good minimizer for linear classification. We partially answer this question by showing that the DPSGD algorithm incurs a privacy error proportional to the maximum norm of the features over all samples, where as DPSGD with a feature preprocessing step incurs a privacy error proportional to the diameter of the features, which can be significantly small compared to the maximum norm of features in several scenarios. Our preliminary experiments show that feature preprocessing helps in practice on standard datasets. Investigating whether these results can be extended to bigger datasets and beyond linear models remain interesting open questions.

---

[6]We perform a modified version of Algorithm 2, which better aligns with existing practical implementations of DPSGD. See Algorithm 3 for details. Note that the resulting algorithm is still a valid $(\epsilon, \delta)$-differentially private.

[7]The reported accuracy using DPSGD in De et al. (2022) is 70.3%. We note that the improvement to 70.6% is due to the fact that we are using the privacy accoutant based on Privacy Loss Distributions (Meiser & Mohammadi, 2018; Sommer et al., 2019; Doroshenko et al., 2022), which leads to a smaller noise multiplier under the same privacy budget.

[8]The reported accuracy doesn't take into account the privacy loss in the hyperparamter tuning phase.

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

## A  PROOF OF THEOREM 3

In the proof, we assume $w_0(2) \leq 0$ and consider $D = D'$. The proof will follow similarly when $w_0(2) \geq 0$ and $D = D''$. We start by proving Lemma 2.

**Proof of Lemma 2:**  Since $\max\{1 - x, 0\}$ is a convex function in $x$, we have

$$
\begin{aligned}
L(w, D) &= \frac{1}{2} \max\{1 - (\mu w(1) + w(2)), 0\} + \frac{1}{2} \max\{1 + (\mu w(1) - w(2)), 0\} \\
&\geq \max\{\frac{1 - (\mu w(1) + w(2)) + 1 + (\mu w(1) - w(2))}{2}, 0\} \\
&= \max\{1 - w(2), 0\}.
\end{aligned}
$$

This implies that

$$
\mathbb{E}\left[L(w, D)\right] - \min_w L(w, D) \geq \mathbb{E}[\max\{1 - w(2), 0\}] \geq \Pr\left(w(2) \leq 0\right).
$$

$\square$

Note that the proof also applies to the final output $\mathcal{A}(D) = \frac{1}{T} \sum_{t=1}^{T} w_t$. And hence to prove Theorem 3, it would be enough to show that.

$$
\Pr\left(\frac{1}{T} \sum_{t=1}^{T} w_t(2) \leq 0\right) = \Omega(1).
$$

In the later analysis, we will focus on the update process of $w_t(2)$. We will prove the following lemma:

**Lemma 8.** *Let $X$ be distributed as $\mathcal{N}\left(\frac{C}{\sqrt{\mu^2 + 1}}, \sigma^2\right)$ with $\sigma^2 = \Theta\left(\frac{C^2}{n^2 \varepsilon^2}\right)$, we have*

$$
\Pr\left(\frac{1}{T} \sum_{t=1}^{T} w_t(2) \leq 0\right) \geq \Pr\left(X \leq 0\right).
$$

Note that Lemma 8 would immediately imply the theorem by Chernoff's inequality. We then turn to proving Lemma 8 below.

For $(x, y) \in D$, we have

$$
\nabla_w \ell(w \cdot x, y) = \begin{cases} \mathbb{1}\left\{\mu w(1) + w(2) < 1\right\} \cdot (-\mu, -1) & \text{if } (x, y) \in D_1, \\ \mathbb{1}\left\{-\mu w(1) + w(2) < 1\right\} \cdot (\mu, -1) & \text{if } (x, y) \in D_{-1}. \end{cases}
$$

Let $B_t$ be the batch of users in iteration $t$ with $|B_t| = B$, the averaged clipped gradient at each iteration will be

$$
\begin{aligned}
\hat{g}_t &= \frac{1}{B} \sum_{(x,y) \in B_t} \mathrm{Clip}(\nabla_w \ell(w_t \cdot x, y), C) \\
&= \frac{1}{B} \min\left\{1, \frac{C}{\sqrt{\mu^2 + 1}}\right\} \cdot \left(\sum_{(x,y) \in B_t \cap D_1} \mathbb{1}\left\{\mu w_t(1) + w_t(2) < 1\right\} \cdot (-\mu, -1)\right) \\
&\quad + \frac{1}{B} \min\left\{1, \frac{C}{\sqrt{\mu^2 + 1}}\right\} \cdot \left(\sum_{(x,y) \in B_t \cap D_{-1}} \mathbb{1}\left\{-\mu w(1) + w(2) < 1\right\} \cdot (\mu, -1)\right).
\end{aligned}
$$

Hence

$$\hat{g}_t(2)$$

$$= -\frac{1}{B} \min\left\{1, \frac{C}{\sqrt{\mu^2+1}}\right\} \left(|B_t \cap D_1| \mathbb{1}\left\{\mu w_t(1) + w_t(2) < 1\right\} + |B_t \cap D_{-1}| \mathbb{1}\left\{-\mu w(1) + w(2) < 1\right\}\right)$$

$$\geq -\frac{1}{B} \min\left\{1, \frac{C}{\sqrt{\mu^2+1}}\right\} \left(|B_t \cap D_1| + |B_t \cap D_{-1}|\right)$$

$$= -\min\left\{1, \frac{C}{\sqrt{\mu^2+1}}\right\}.$$

Standard privacy analysis in DP-SGD (the setting of parameters in Algorithm 1 used in Bassily et al. (2019)) sets $B \geq \max\{n\sqrt{\varepsilon/4T}, 1\}$ and adds a Gaussian noise $\mathcal{N}(0, \sigma^2 \mathbb{1}_2)$ with $\sigma^2 = \frac{8TC^2 \log(1/\delta)}{n^2 \varepsilon^2}$[9] to the averaged clipped gradient. Let $N_t$ denote the noise added at iteraction $t$, we have

$$w_{t+1}(2) \leq w_t(2) + \eta\left(\min\left\{1, \frac{C}{\sqrt{\mu^2+1}}\right\} + N_t(2)\right).$$

Hence we have

$$w_t(2) \leq w_0(2) + \eta \sum_{i=0}^{t-1}\left(\min\left\{1, \frac{C}{\sqrt{\mu^2+1}}\right\} + N_i(2)\right)$$

$$\leq \eta\left(t \min\left\{1, \frac{C}{\sqrt{\mu^2+1}}\right\} + \sum_{i=0}^{t-1} N_i(2)\right)$$

This implies:

$$\frac{1}{T} \sum_{t=1}^{T} w_t(2) \leq \eta\left(\frac{T+1}{2} \min\left\{1, \frac{C}{\sqrt{\mu^2+1}}\right\} + \frac{1}{T} \sum_{t=0}^{T-1}(T-t) N_i(2)\right).$$

Note that the right hand side is a Gaussian distributed random variable with mean $\eta \frac{T+1}{2} \min\left\{1, \frac{C}{\sqrt{\mu^2+1}}\right\}$ and variance $\eta^2 \frac{4(T+1)(2T+1)C^2 \log(1/\delta)}{3n^2\varepsilon^2}$. Hence

$$\Pr\left(\frac{1}{T} \sum_{t=1}^{T} w_t(2) \leq 0\right)$$

$$\leq \Pr\left(\mathcal{N}\left(\eta\frac{T+1}{2} \min\left\{1, \frac{C}{\sqrt{\mu^2+1}}\right\}, \eta^2\frac{4(T+1)(2T+1)C^2 \log(1/\delta)}{3n^2\varepsilon^2}\right) \leq 0\right)$$

$$\leq \Pr\left(\mathcal{N}\left(\min\left\{1, \frac{C}{\sqrt{\mu^2+1}}\right\}, \frac{16(2T+1)C^2 \log(1/\delta)}{3(T+1)n^2\varepsilon^2}\right) \leq 0\right)$$

This completes the proof of Lemma 8.

# B   MISSING PROOFS IN SECTION 5

## B.1   PROOF OF LEMMA 3

Let $\varepsilon \leq \log(3/\delta)$ and let $\rho = \frac{\varepsilon^2}{49\log(3/\delta)}$. Let $\alpha = \frac{R}{n^2}$ and $\gamma(\zeta) = \sqrt{\frac{1}{\rho} \cdot \log(2d^{3/2}n^2) \log\frac{(d\log(2d^{3/2}n^2))}{\zeta}}$. If $n \geq c \cdot \gamma(\zeta)\sqrt{d}$ (for a sufficiently large constant $c$), then

---

[9]A different privacy accounting method might give slightly improved result. But overall we will have $\sigma^2 = \Theta\left(\frac{8TC^2 \log(1/\delta)}{n^2\varepsilon^2}\right)$, and our claim in Lemma 8 won't change up to constant factors.

by Theorem 3 and Remark 1 from Huang et al. (2021), there exists a $\rho$-zCDP algorithm that outputs $\hat{\mu}$ such that

$$\|\mu - \hat{\mu}\|_2 = O\left(\sqrt{\frac{d}{\rho}} + \gamma(\zeta)\right) \cdot \frac{\text{diam}(D)\sqrt{\log(nd/\zeta)}}{n} + \frac{R}{n^2},$$

with probability at least $1 - \zeta$. Let $f(\zeta) = O\left(\sqrt{\frac{d}{\rho}} + \gamma(\zeta)\right) \cdot \frac{\text{diam}(D)\sqrt{\log(nd/\zeta)}}{n} + \frac{R}{n^2}$. Hence,

$$\begin{aligned}
\mathbb{E}[\|\mu - \hat{\mu}\|_2] &= \mathbb{E}[\|\mu - \hat{\mu}\|_2 \mathbb{1}_{\|\mu - \hat{\mu}\|_2 \leq f(\zeta)}] + \mathbb{E}[\|\mu - \hat{\mu}\|_2 \mathbb{1}_{\|\mu - \hat{\mu}\|_2 > f(\zeta)}] \\
&\leq \mathbb{E}[f(\zeta)] + \mathbb{E}[2R\mathbb{1}_{\|\mu - \hat{\mu}\|_2 > f(\zeta)}] \\
&\leq f(\zeta) + 2R\Pr(\|\mu - \hat{\mu}\|_2 > f(\zeta)) \\
&\leq f(\zeta) + 2R\zeta,
\end{aligned}$$

where the first inequality follows by observing that both $\mu$ and $\hat{\mu}$ have norm at most $R$. Setting $\zeta = \frac{1}{n^2}$ yields,

$$\begin{aligned}
\mathbb{E}[\|\mu - \hat{\mu}\|_2] &\leq O\left(\sqrt{\frac{d}{\rho}} + \gamma(1/n^2)\right) \cdot \frac{\text{diam}(D)\sqrt{\log(n^5 d)}}{n} + \frac{2R}{n^2} \\
&\leq O\left(\sqrt{\frac{d}{\rho}} + \frac{\log(dn)}{\sqrt{\rho}}\right) \cdot \frac{\text{diam}(D)\sqrt{\log(n^5 d)}}{n} + \frac{2R}{n^2} \\
&\leq O\left(\sqrt{d} + \log(dn)\right) \cdot \frac{\text{diam}(D)\sqrt{\log(n^5 d)}\sqrt{\log(1/\delta)}}{n\epsilon} + \frac{2R}{n^2} \\
&\leq \text{diam}(D) + \frac{2R}{n^2},
\end{aligned}$$

where the last inequality follows based on condition on $n$. Note that $n \geq c \cdot \gamma(1/n^2)\sqrt{d}$ based on the assumption in the lemma. By (Kairouz et al., 2021, Lemma 4), this algorithm is also $(\epsilon/3, \delta/3)$-differentially private and hence the lemma.

## B.2 PROOF OF LEMMA 4

Throughout the proof, we assume $\hat{\mu}$ is fixed and prove the statement for any $\hat{\mu}$. Applying (Dick et al., 2023, Theorem 2) with $a = 0$, $b = R$, privacy budget $\epsilon/3$, error probability $\zeta$, $\alpha = R/n^2$, $r = n - \frac{100}{\epsilon}\log n$, yields that the output $\hat{\tau}$ satisfies the following: with probability at least $1 - \zeta$, there exists a $\tau'$ such that

$$|\hat{\tau} - \tau'| \leq \frac{R}{n^2},$$

and there are at most $\frac{100}{\epsilon}\log n + \frac{6}{\epsilon}\log\frac{R}{\zeta \cdot R/n^2} = \frac{100}{\epsilon}\log n + \frac{6}{\epsilon}\log\frac{n^2}{\zeta}$ points with $\|x_i - \hat{\mu}\|_2$ more than $\tau'$ and at least $\frac{100}{\epsilon}\log n - \frac{6}{\epsilon}\log\frac{n^2}{\zeta}$ points with $\|x_i - \hat{\mu}\|_2$ less than $\tau'$. Let $\tau = \hat{\tau} + \frac{R}{n^2}$, then $\tau$ satisfies the following: there are most $\frac{100}{\epsilon}\log n + \frac{6}{\epsilon}\log\frac{n^2}{\zeta}$ points with $\|x_i - \hat{\mu}\|_2$ more than $\tau$ and at least $\frac{100}{\epsilon}\log n - \frac{6}{\epsilon}\log\frac{n^2}{\zeta}$ points with $\|x_i - \hat{\mu}\|_2$ greater than $\tau$. Let $S$ be the set of points with norm larger than $\tau$. Let $s_0$ be a value we set later.

$$\begin{aligned}
\mathbb{E}[|S|] &= \mathbb{E}[|S|\mathbb{1}_{|S|\leq s_0}] + \mathbb{E}[|S|\mathbb{1}_{|S|\geq s_0}] \\
&\leq \mathbb{E}[s_0 \mathbb{1}_{|S|\leq s_0}] + \mathbb{E}[n\mathbb{1}_{|S|\geq s_0}] \\
&\leq s_0 + n\Pr(|S| \geq s_0).
\end{aligned}$$

Setting $\zeta = \frac{1}{n^2}$ and $s_0 = \frac{100}{\epsilon}\log n + \frac{24}{\epsilon}\log n$ yields that

$$\mathbb{E}[|S|] \leq \frac{1}{n} + \frac{124}{\epsilon}\log n \leq \frac{125}{\epsilon}\log n.$$

To prove the other result, let $s_1$ be a value which we set later. Observe that

$$\mathbb{E}[\tau] \leq \mathbb{E}\left[\left(\hat{\tau} + \frac{R}{n^2}\right)\right]$$

$$\leq \mathbb{E}[\hat{\tau}] + \frac{R}{n^2}$$

$$\leq \mathbb{E}[\hat{\tau} 1_{|S| \leq s_1}] + 2\mathbb{E}[\hat{\tau} 1_{|S| \geq s_1}] + \frac{R}{n^2}$$

$$\leq R\mathbb{E}[1_{|S| \leq s_1}] + \mathbb{E}[\max_i \|x_i - \hat{\mu}\|_2 1_{|S| \geq s_1}] + \frac{R}{n^2}$$

$$\leq R\Pr(1_{|S| \leq s_1}) + \max_i \|x_i - \hat{\mu}\|_2 + \frac{R}{n^2}.$$

Setting $s_1 = \frac{100}{\epsilon} \log n - \frac{24}{\epsilon} \log n$ yields that

$$\mathbb{E}[\tau] \leq \max_i \|x_i - \hat{\mu}\|_2 + \frac{2R}{n^2}.$$

## B.3 PROOF OF LEMMA 6

For the augmented dataset $D'$,

$$\sum_i x_i'(x_i')^\top = \begin{pmatrix} \sum_i (x_i - \hat{\mu})(x_i - \hat{\mu})^\top & \tau \sum_i (x_i - \hat{\mu}) \\ \tau \sum_i (x_i - \hat{\mu}) & n\tau^2 \end{pmatrix}$$

$$= \begin{pmatrix} \sum_i x_i x_i^\top + \hat{\mu}\hat{\mu}^\top - n\mu\hat{\mu}^\top - n\hat{\mu}\mu^\top & n\tau(\mu - \hat{\mu}) \\ n\tau(\mu - \hat{\mu}) & n\tau^2 \end{pmatrix}$$

where $\mu = \frac{1}{n} \sum_i x_i$. Hence we have $\text{rank}(M(D')) \leq \text{rank}(M(D)) + \text{rank}(\mu\hat{\mu}^\top) + \text{rank}(\hat{\mu}\mu^\top) + \text{rank}(\hat{\mu}\hat{\mu}^\top) + 2$, where we use the fact that adding one column/row will at most increase the rank of a matrix by one.

To prove the second result, note that by triangle inequality,

$$\|\theta'^*\|_2 \tau \leq \|w^{*'}\|_2 \tau + |b^{*'}|\tau$$

$$\leq \|w^*\|_2 \tau + |b^{*'}|\tau,$$

where the second inequality follows from Lemma 5. We now bound the second term in the right hand side of the above equation. Observe that by Assumption 1, there exists $x_i$ and $x_j$ such that

$$w^* x_i + b^* \tau \geq 0 \text{ and } w^* x_j + b^* \tau \leq 0,$$

and hence

$$-w^* x_i \leq b^* \tau \leq -w^* x_j.$$

Hence,

$$w^* \cdot (\hat{\mu} - x_i) \leq b^* \tau + w^* \cdot \hat{\mu} \leq w^* \cdot (\hat{\mu} - x_j),$$

Hence by Lemma 5,

$$|b^{*'}|\tau = |b^* + w^* \cdot \hat{\mu}|$$

$$\leq \max_{x_i \in D} |w^* \cdot (\hat{\mu} - x_i)|$$

$$\leq \|w^*\|_2 \max_{x_i \in D} \|(\hat{\mu} - x_i)\|_2 \tag{5}$$

Combining the above two equations yield,

$$\|\theta'^*\|_2 \tau \leq \|w^*\|_2 \tau + \|w^*\|_2 \max_{x_i \in D} \|(\hat{\mu} - x_i)\|_2$$

$$\leq \|\theta^*\|_2 \tau + \|\theta^*\|_2 \max_{x_i \in D} \|(\hat{\mu} - x_i)\|_2$$

Hence by Lemma 4,

$$\mathbb{E}[\|\theta'^*\|_2 \tau | \hat{\mu}] \leq \|\theta^*\|_2 \mathbb{E}[\tau | \hat{\mu}] + \|\theta^*\|_2 \max_{x_i \in D} \|(\hat{\mu} - x_i)\|_2$$

$$\leq 2\|\theta^*\|_2 \max_{x_i \in D} \|(\hat{\mu} - x_i)\|_2 + \frac{2\|\theta^*\|_2 R}{n^2}.$$

## B.4 PROOF OF LEMMA 7

If $v \in \text{span}(U(D'))$, then $v^\top \sum_i x_i' x_i'^\top v > 0$. Hence $\sum_i \|x_i' v\|_2^2 > 0$, which implies $\sum_i \|x_i'' v\|_2^2 > 0$ and hence $v^\top \sum_i x_i'' x_i''^\top v > 0$ and hence $v \in \text{span}(U(D''))$. Similar analysis from $U(D'')$ to $U(D')$ yields that $\text{span}(U(D')) = \text{span}(U(D''))$ and hence $\text{rank}(M(D')) = \text{rank}(M(D''))$.

## B.5 PROOF OF THEOREM 1

The proof of privacy guarantee follows directly from the composition theorem (Dwork et al., 2014, Theorem 3.16). Note that for simplicity, we used composition theorem here. However in experiments, we use the recent technique of PLD accountant (Meiser & Mohammadi, 2018; Sommer et al., 2019; Doroshenko et al., 2022) implemented in Google (2018) to compute privary paramters $\epsilon$ and $\delta$.

We now focus on the utility guarantee.

Let $\theta^*, \theta'^*, \theta''^*$ be the empirical risk minimizers of $D, D', D''$ respectively. Let $M(D), M(D'), M(D'')$ be the design matrices of datasets $D, D', D''$ respectively. Due to feature-clipping, the gradient for any $(x'', y'') \in D''$ and $\theta$, the gradient of $\ell(h(\theta, x''), y'')$ is upper bounded by

$$G\|x''\|_2 \leq \sqrt{2}G\tau.$$

Hence, by (Song et al., 2021a, Theorem 3.1)[10] and Lemma 6,

$$\mathbb{E}[L(\theta''_{\text{prv}}, D'')|D''] - L(\theta'^*, D'') \leq 2\sqrt{2}G\tau\|\theta'^*\|_{M(D'')} \left( \frac{\sqrt{\text{rank}(M(D''))\log(3/\delta)}}{n\epsilon} \right)$$

$$\leq 10\sqrt{2}G\tau\|\theta'^*\|_2 \left( \frac{\sqrt{\text{rank}(M)\log(3/\delta)}}{n\epsilon} \right). \quad (6)$$

By Lemma 6 and Equation 3,

$$\mathbb{E}[\tau\|\theta'^*\|_2] = \mathbb{E}[\mathbb{E}[\tau\|\theta'^*\|_2|\hat{\mu}]$$

$$\leq 2\|\theta^*\|_2 \mathbb{E}[\max_{x_i \in D} \|(\hat{\mu} - x_i)\|_2] + \frac{w\|\theta^*\|_2 R}{n^2}$$

$$\leq 4\|\theta^*\|_2 \text{diam}(D) + \frac{6\|\theta^*\|_2 R}{n^2}.$$

Substituting the above equation in equation 6 yields and taking expectation over $D''$ yields

$$\mathbb{E}[L(\theta''_{\text{prv}}, D'')] - \mathbb{E}[L(\theta'^*, D'')] \leq 60\sqrt{2}G\|\theta^*\|_2 \left( \text{diam}(D) + \frac{R}{n^2} \right) \left( \frac{\sqrt{\text{rank}(M)\log(3/\delta)}}{n\epsilon} \right).$$

---

[10]Note that(Song et al., 2021a, Theorem 3.1) states the bound in terms of empirical risk minimizer $\theta''^*$, but the bound holds for any $\theta$.

Let $S$ be the set of samples which differ between $D'$ and $D''$. By Lemma 6,

$$\mathbb{E}[L(\theta'^*, D'')] \leq \mathbb{E}[L(\theta'^*, D')] + \mathbb{E}\left[\frac{1}{n}\sum_{i \in S} \ell(h(\theta'^*, x_i''), y_i) - \ell(h(\theta'^*, x_i'), y_i)\right]$$

$$\leq \mathbb{E}[L(\theta'^*, D')] + \mathbb{E}\left[\frac{1}{n}\sum_{i \in S} |\ell(h(\theta'^*, x_i''), y_i) - \ell(h(\theta'^*, x_i'), y_i)|\right]$$

$$\leq \mathbb{E}[L(\theta'^*, D')] + \mathbb{E}\left[\frac{1}{n}\sum_{i \in S} G|h(\theta'^*, x_i'') - h(\theta'^*, x_i')|\right]$$

$$\leq \mathbb{E}[L(\theta'^*, D')] + \mathbb{E}\left[\frac{1}{n}\sum_{i \in S} G\left(|w^{*'} \cdot x_i'| + |b^{*'}|\tau\right)\right]$$

$$\overset{(a)}{\leq} \mathbb{E}[L(\theta'^*, D')] + \mathbb{E}\left[\frac{1}{n}\sum_{i \in S} 2G \max_j |w^{*'} \cdot (x_j - \hat{\mu})|\right]$$

$$\leq \mathbb{E}[L(\theta'^*, D')] + \mathbb{E}\left[\frac{1}{n}\sum_{i \in S} 2G\|w^{*'}\|_2 \cdot (\mathrm{diam}(D) + \|\hat{\mu} - \mu\|_2)\right]$$

$$\overset{(b)}{=} \mathbb{E}[L(\theta'^*, D')] + \frac{2G}{n}\mathbb{E}\left[\sum_{i \in S} \|w^*\|_2 \cdot (\mathrm{diam}(D) + \|\hat{\mu} - \mu\|_2)\right]$$

$$\leq \mathbb{E}[L(\theta'^*, D')] + \frac{2G\|w^*\|_2}{n}\mathbb{E}\left[\mathbb{E}[|S| \mid \hat{\mu}] \cdot (\mathrm{diam}(D) + \|\hat{\mu} - \mu\|_2)\right]$$

$$\overset{(c)}{\leq} \mathbb{E}[L(\theta'^*, D')] + \frac{250G \log n \|w^*\|_2}{n\epsilon}\mathbb{E}\left[(\mathrm{diam}(D) + \|\hat{\mu} - \mu\|_2)\right]$$

$$\overset{(d)}{\leq} \mathbb{E}[L(\theta'^*, D')] + \frac{250G \log n \|w^*\|_2}{n\epsilon}\left(2\mathrm{diam}(D) + \frac{2R}{n^2}\right)$$

$$\leq \mathbb{E}[L(\theta'^*, D')] + \frac{500G \log n \|\theta^*\|_2}{n\epsilon} \cdot \left(2\mathrm{diam}(D) + \frac{2R}{n^2}\right).$$

where $(a)$ follows from Eq. (5), $(b)$ follows from Lemma 5, $(c)$ follows from Lemma 4, and $(d)$ follows from Lemma 3. We now bound in the other direction.

$$\mathbb{E}[L(\theta''_{\mathrm{prv}}, D'')] = \mathbb{E}[L(\theta''_{\mathrm{prv}}, D')] + \frac{1}{n}\mathbb{E}[\sum_{i \in S} \ell(h(\theta''_{\mathrm{prv}}, x_i''), y_i) - \ell(h(\theta''_{\mathrm{prv}}, x_i'), y_i)].$$

Notice that if $y_i\theta''_{\mathrm{prv}} \cdot x_i'' \leq 0$, then $y_i\theta''_{\mathrm{prv}} \cdot x_i' \leq y_i\theta''_{\mathrm{prv}} \cdot x_i'' \leq 0$. Hence,

$$\mathbb{E}[L(\theta''_{\mathrm{prv}}, D'')] \geq \mathbb{E}[L(\theta''_{\mathrm{prv}}, D')] + \frac{1}{n}\mathbb{E}\left[\sum_{i \in S: y_i\theta''_{\mathrm{prv}} \cdot x_i'' > 0} \ell(h(\theta''_{\mathrm{prv}}, x_i''), y_i) - \ell(h(\theta''_{\mathrm{prv}}, x_i'), y_i)\right]$$

$$\geq \mathbb{E}[L(\theta''_{\mathrm{prv}}, D')] - \frac{1}{n}\mathbb{E}\left[\sum_{i \in S: y_i w''_{\mathrm{prv}} \cdot x_i'' > 0} \ell(h(\theta''_{\mathrm{prv}}, x_i'), y_i)\right]$$

$$\geq \mathbb{E}[L(\theta''_{\mathrm{prv}}, D')] - \frac{1}{n}\mathbb{E}[|S|\phi(0)]$$

$$\geq \mathbb{E}[L(\theta''_{\mathrm{prv}}, D')] - \frac{125 \log n}{n\epsilon}\phi(0),$$

where the last inequality follows from Lemma 4. Hence we have

$$\mathbb{E}[L(\theta''_{\mathrm{prv}}, D')] - L(\theta^{*'}, D') \leq \frac{500G \log n \|\theta^*\|_2}{n\epsilon} \cdot \left(2\mathrm{diam}(D) + \frac{2R}{n^2}\right) + \frac{125 \log n}{n\epsilon}\phi(0).$$

By Lemma 5, we have $\mathbb{E}[L(\theta''_{\mathrm{prv}}, D')] - L(\theta^{*'}, D') = \mathbb{E}[L(\theta_{\mathrm{prv}}, D)] - L(\theta^*, D)$. This yields that

$$\mathbb{E}[L(\theta_{\mathrm{prv}}, D)] - L(\theta^*, D) \leq cG\|\theta^*\|_2\left(\mathrm{diam}(D) + \frac{R}{n^2}\right)\left(\frac{\sqrt{\mathrm{rank}(M)\log(3/\delta)} + \log n}{n\epsilon}\right) + \frac{c\phi(0)}{n\epsilon}\log n,$$

---

**Algorithm 3** Modefied version of DPSGD with feature preprocessing (DPSGD-F*).

---

**Input:** Input: Dataset $D = \{(x_1, y_1), (x_2, y_2), \ldots, (x_n, y_n)\}$ of $n$ points; overall privacy budget $\varepsilon, \delta$; privacy budget for the preprocessing step $\varepsilon_F \in (0, \varepsilon)$; feature norm bound $C_F$; gradient clipping norm $C_G$; step size $\eta$, number of iterations $T$; batch size $B$.

1: **Private mean estimation.** Compute the noise multiplier $\sigma_F$ for the Gaussian mechanism with $\varepsilon_F$ and $\delta$ using analytical callibration for Gaussian mechanism Balle & Wang (2018), and compute

$$\hat{\mu} = \frac{1}{n} \sum_{i=1}^{n} x_i + \mathcal{N}(0, \sigma_F C_f \mathbb{I}_d).$$

2: **Translate the features.** Preprocess the dataset and obtain $D' = \{(x'_1, y_1), \ldots, (x'_n, y_n)\}$, where

$$x'_i = x_i - \hat{\mu}.$$

3: **DPSGD with preprocessed features.** Compute the privacy budget $\varepsilon'$ for the DPSGD phase using the PLD accountant in Google (2018) by setting $(\varepsilon, \delta)$ as the overall privacy budget for the composition of both the mean estimation phase and DPSGD phase. Get an an approximate minimizer $\theta'_{\text{prv}}$ of $L(\theta, D')$ using the DPSGD algorithm with privacy budget $(\varepsilon', \delta)$.

4: **Return** $\theta_{\text{prv}} = (w_{\text{prv}}, b_{\text{prv}})$, where $w_{\text{prv}} = \theta'_{\text{prv}}[1 : d]$ and $b_{\text{prv}} = \theta'_{\text{prv}}[d + 1] - \hat{\mu} \cdot w_{\text{prv}}$.

---

for a sufficient large constant $c$. The theorem follows by observing that for the minimum norm solution $\theta^*$, $\|\theta^*\|_2 = \|\theta^*\|_M$.

## C ADDITIONAL EXPERIMENT DETAILS.

In this section, we discuss the detailed implementation and parameter settings used in our experiments. We implement all algorithms and experiments using the open-source JAX (Bradbury et al., 2018) library. For privacy accounting, we use the PLD accountant implemented in Tensorflow Privacy (Google, 2018). We first describe the implementation details of our experiments.

**Feature normalization.** For all three datasets, we consider their feature-normalized version, where we normalize the feature vectors to a fixed norm of $C_F$ by the transformation below

$$x'_i = x_i \cdot \frac{C_F}{\|x_i\|_2}.$$

We treat $C_F$ as a fixed constant, and hence this normalization step doesn't result in any privacy loss about the dataset.

**Modifications to DPSGD-F.** The version of DPSGD-F listed in Algorithm 2 is mainly presented for ease of strict theoretical analysis since existing analysis on DPSGD mainly assumse that the gradients are bounded and no clipping is needed during the DPSGD phase (Bassily et al., 2019; Feldman et al., 2020; Bassily et al., 2020; 2021b;a; Song et al., 2021b; Arora et al., 2022). However, state-of-the-art results from DPSGD usually uses the clipping version of DPSGD where no gradient norm bound is assumed (De et al., 2022; Ghalebikesabi et al., 2023).

In our experiments, instead of directly implementing Algorithm 2, we perform the following modifications in the experiments. The implemented version of the algortihm is described in Algorithm 3.

1. For the feature preprocessing step, when computing the mean, instead of using the adaptive algorithm in Lemma 3, the algorithm directly applies Gaussian mechanism to the empirical mean since each feature is normalized to a fixed norm. We can use Gaussian mechanism for mean estimation since now the $\ell_2$ sensitivity is bounded.
2. We skip the private quantile estimation and feature clipping step. Instead, we only shift the features by the privately estimated mean and treat it as the input to the DPSGD phase. We then perform $\ell_2$-norm clipping on the gradients as in DPSGD.

The privacy guarantee of Algorithm 3 directly follows from the use of PLD accountant (Meiser & Mohammadi, 2018; Sommer et al., 2019).

**Lemma 9.** *The output of Algorithm 3 is $(\epsilon, \delta)$-differentially private.*

Each each combination of (ALGORITHM, $\varepsilon$, DATASET), we fix the clipping norm $C_G$ to be 1 as suggested in De et al. (2022), and perform a grid search over $C_F$, BATCH SIZE, LEARNING RATE, and NUMBER OF EPOCHS from the list below and report the best-achieved accuracy, similar to De et al. (2022).

1. $C_F$: 1, 10, 100, 1000.
2. BATCH SIZE: 256, 512, 1024, 2048, 4096, 8192, 16384.
3. LEARNING RATE: 0.03125, 0.0625, 0.125, 0.25, 0.5, 1, 2, 4, 8, 16.
4. NUMBER OF EPOCHS: 20, 40, 80, 160, 320.

For DPSGD-F, we further add a search dimension over the privacy budget $\varepsilon_F$ for the feature pre-processing step. And the list of candidates is chosen to be $[0.02, 0.05, 0.1, 0.15, 0.2]$. In fact, for all experiments, the best accuracy is achieved at either $\varepsilon_F = 0.02$ or $\varepsilon_F = 0.05$. This suggests that the feature preprocessing step actually doesn't use much of the privacy budget to improve the final accuracy.

**Remark on different concentration measures of the datasets.** For the three considered datasets, below we compute a few distance measures that characterize how the dataset $D$ is concentrated, listed in Table 2. It appears that compared to $\mathrm{diam}(D)$, $\frac{1}{n} \sum_i \|x_i - \mu\|_2$ is a more indicative measure on the performance improvement from DPSGD-F compared to DPSGD. The smaller $\frac{1}{n} \sum_i \|x_i - \mu\|_2$, the more improvement DPSGD-F has over DPSGD. This might be due to the fact that we are considering the gradient clipping version of DPSGD (Algorithm 3) instead of the feature clipping version (Algorithm 2) considered in the theoretical analysis. Better understanding of this phenomena is an interesting future direction.

Table 2: Different concentration measures of the datasets. Features in datasets are normalized to be of unit norm. $\mu$ denotes the mean of the dataset.

| Dataset | $\mathrm{diam}(D)$ | $\max_i \|x_i - \mu\|_2$ | $\frac{1}{n} \sum_i \|x_i - \mu\|_2$ |
|---|---|---|---|
| CIFAR-100 (pretrained) † | 1.56 | 1.10 | 0.9 |
| MNIST | 1.41 | 1.03 | 0.77 |
| FMNIST | 1.41 | 1.10 | 0.62 |

