# OpenReview forum: "The importance of feature preprocessing for differentially private linear optimization"
_ICLR.cc/2024/Conference — ICLR 2024 poster_

### Official Review · Reviewer_4bdB · 2023-10-24

**Soundness:** 3 good
**Presentation:** 3 good
**Contribution:** 2 fair
**Rating:** 6
**Confidence:** 3

**Summary:**

Differential privacy (DP) has been applied to many machine/deep learning tasks to keep privacy of dataset. DPSGD is one of the influential work to train private ML models. Although some theoretical works proves its convexity properties and leads the trend, DPSGD may not work well on raw features in image classification tasks. Beyond the DPSGD, this work demonstrates the feature preprocessing is necessary to optimize the performance of DPSGD by proposing a new algorithm - DPSGD-F as a combination of DPSGD and feature preprocessing.

**Strengths:**

1. Based on the proposed algorithm, this work does not only provide empirical results on specific image tasks, but it also provides theoretical guarantees on it step by step. It is really good to connect theory and practice in the research.
2. The counter-example of DPSGD in Section 4 is innovative and looks solid to investigates its limitation seldom considered by other works (to my knowledge).

**Weaknesses:**

1. The study subject - linear model - is relatively simple and less representative compared to SOTA ML models, especially for image classification task.
2. Except the Fashion-MNIST dataset (released in recent few years), original MNIST and CIFAR-100 datasets are relatively outdated compared to other image datasets.

**Questions:**

1. What is Omega tilde in Theorem 1?
2. Can you explain more about what the “uninformative” means in the Theorem 3?
3. For lemmas in Section 5, can you provide high level ideas of proofing for each one (probably one or two sentence per lemma) like you did in the Section 4 to make reader clearer about the correctness?

---

> ### Author Response · Authors · 2023-11-19
>
> We thank the reviewer for acknowledging that our work is a good connection between theory and practice. Below we address the reviewer’s questions on our work.
>
> ***[The study subject - linear model - is relatively simple and less representative compared to SOTA ML models.]***
> ***[Except the Fashion-MNIST dataset (released in recent few years), original MNIST and
> CIFAR-100 datasets are relatively outdated compared to other image datasets.]***
>
> We believe the main contribution of our work is to formally prove that feature preprocessing can help DPSGD-based algorithms to obtain better optimization guarantees. While the considered model and datasets are simple, we believe these will serve as important first steps to study more complex SOTA models and modern datasets.
>
> On the other hand, while without privacy, these tasks have been studied a lot and SOTA results have been advanced over the years, these tasks still remain challenging tasks for private image classification. For example, to the best of our knowledge, no good classification model has been privately trained from scratch on CIFAR-100. Current results solve classification on CIFAR-100 by private finetuning using pre-trained models on larger datasets (e.g. ImageNet1K). For this, [De et al 2022] showed that fine tuning the last layer of a WRN network achieves an accuracy of 70.3% on CIFAR-100 at $\varepsilon = 1$. Note that fine tuning the last layer of an advanced model like WRN is the same as training a linear model. With our technique, we are able to improve the accuracy to 71.6%, which is close to the current SOTA result of 71.9% [Bu et al 22] for the task of fine tuning on CIFAR-100 with pretrained models from ImageNet1K. This shows that for private training of models, linear models may not be just a simplified model, but rather a useful model in practice, especially given its equivalence to private fine tuning of the last layer of any neural models.
>
> ***[Omega tilde in Theorem 1]***
>
> Tilde was used to hide logarithmic factors in the big O expression. In the updated draft, we have removed the tilde notation and written out the specific log factors. Now the term inside Omega represents an up-to-constant bound on the required number of samples to obtain the result in Theorem 1.
>
> ***[Can you explain more about what the “uninformative” means in Theorem 3?]***
>
> The term "uninformative initialization" is mainly used to require that the initialization should be not be chosen depending on the input dataset so that it will benefit the optimization for specific datasets instead of all. In the case of $D'$ in Theorem 3, we mean $w_0 = (w_0(1), w_0(2))$ should satisfy that $w0(2) ≤ 0$ since the optimal $w^*$ for $D'$ has $w^*(2) > 0$. Similarly for $D''$ we require $w_0(2) ≥ 0$.
>
> ***[High-level ideas on correctness for lemmas in Section 5.]***
>
> Thanks for the great suggestion. Lemma 3 and 4 mainly come from prior work, and the results are direct corollaries. Proof of Lemma 5 is short and provided in the section. We have added a high-level explanation on why Lemma 6 and Lemma 7 hold in the update draft (marked in red).

---

### Official Review · Reviewer_m2H3 · 2023-10-30

**Soundness:** 3 good
**Presentation:** 3 good
**Contribution:** 2 fair
**Rating:** 6
**Confidence:** 3

**Summary:**

The paper investigated the significance of feature processing in differentially private linear models. The authors introduced an algorithm named DPSGD-F, which incorporates a preprocessing procedure for feature clipping within DP-SGD. The authors theoretically demonstrated the relationship between the excess risk and the feature diameter. Furthermore, their empirical results suggested that DPSGD-F outperforms DP-SGD on datasets such as MNIST, FMNIST, and CIFAR-100 (pretrained).

**Strengths:**

Strengths:

1. The authors introduced a novel variant of DPSGD known as DPSGD-F, which empirically outperforms DPSGD on particular datasets, and they supported the proposed algorithm with theoretical guarantees.

2. The authors presented a counterexample illustrating situations where DP-SGD does not converge.

**Weaknesses:**

Weaknesses:

1. I have several concerns regarding the motivation and theoretical results, which might undermine the paper's conclusion.  Below are some of my concerns about the counter example in Section 4 and Theorem 1 in Section 3.

    a) In the counterexample, the condition $\mu > n\epsilon$ is required. However, if I understand Theorem 1 correctly, the second term in Theorem 1 becomes a constant when $\mu > n\epsilon$ since $R = \sqrt{\mu^2 + 1}$. Consequently, this counterexample may not convincingly imply that DPSGD-F is superior to DPSGD. It is recommended to address this concern during the rebuttal session.

    b) The authors use this counterexample to emphasize the importance of feature diameter. Nevertheless, it appears that it's the angle between features for different classes, rather than the diameter itself, that is crucial. Particularly, when $\mu$ is sufficiently large, the features for different classes become nearly identical because, in comparison to $\mu e_1$, $e_2$ can be neglected. To make the counter example more convincing, please consider whether the example $x = e_1 + \frac{1}{\mu} e_2$ for $y=1$ and $x = e_1 - \frac{1}{\mu} e_2$ for $y=-1$ remains consistent when $\mu$ is sufficiently large.

    c) The upper bound in Theorem 1 is dimension-dependent since the authors required that $n\geq O(\sqrt{d}/\epsilon).

2. Given that the counterexample doesn't fully convince the reviewer, it is recommended that the authors conduct an empirical evaluation of DPSGD-F on a wider range of datasets. Specifically, it would be beneficial to compare the results with Table 1 in De et al.'s work (https://arxiv.org/pdf/2204.13650.pdf) using various choices of pretrained and finetuned datasets for further validation.

**Questions:**

Please reference the weakness section.

---

> ### Author Response · Authors · 2023-11-19
>
> We thank the reviewer for agreeing to the novelty of the work and raising important questions regarding the result. Below we address the concerns raised by the reviewer.
>
> ***[The performance of DPSGD-F (Theorem 1) on the counterexample.]***
>
> Thank the reviewer for pointing out that Theorem 1 in the submitted manuscript can only show that DPSGD-F achieves a constant error for the counterexample in Section 4. After a careful examination of our result, we are able to obtain an improved performance guarantee of our proposed algorithm DPSGD-F with a more involved analysis. We have marked the updated result (Theorem 1) and analysis (Appendix B.5) in red in the updated draft. In short, we are able to improve the dependence on $R$ and $diam(D)$ to roughly $(diam(D) + R/n^2) \cdot \sqrt{rank(M)}/(n\varepsilon)$, ignoring other constants that don’t depend on $R$ and $diam(D)$. In this form, it can be seen that DPSGD-F strictly improves upon DPSGD by achieving a $o(1)$ error when $\sqrt{\mu} \ll n < \mu/\varepsilon$.
>
> ***[Whether the counterexample still holds when $x_1 = e_1 + 1/\mu e_2$ and $x_2 = e_1 - 1/\mu e_2$.]***
>
> The result holds under any rescaling of $x_1$ and $x_2$. To see this, the standard deviation of the noise for DP-SGD scales linearly with the norm of $x$ and hence any scaling won’t change the relative scale between the added noise and the gradient vector.
>
> ***[The upper bound in Theorem 1 is dimension-dependent since the authors required that $n\geq O(\sqrt{d}/\varepsilon)$.]***
>
> We agree that this is a limitation of our work. This constraint mainly comes from the requirement on $n$ for the private mean estimation step. To the best of our knowledge, the requirement is needed for state-of-the-art mean estimation algorithms. We leave studying whether the constraint can be improved as a future direction.
>
> ***[Given that the counterexample doesn't fully convince the reviewer, it is recommended that the authors conduct an empirical evaluation of DPSGD-F on a wider range of datasets]***
>
> We hope the improved performance guarantee and analysis on the algorithm resolves the reviewer's concern on the counterexample. This work is mainly a theoretical investigation of the importance of feature preprocessing for private optimization. We believe the theoretical result (especially the improved version in the updated draft) and the set of experiments already give decent evidence on our claim. Moreover, some of the experiments already demonstrate the potential of our approach for challenging classification problems. For example, for the task of privately finetuning the last layer of a pretrained model (using ImageNet1K) on CIFAR-100 under $\varepsilon=1$, our result improves the previous best accuracy from 70.6% (De et al., 2022) to 71.6%. This being said, we agree that conducting more extensive experiments would make the empirical result more convincing. We could add more experiments in the final version if the reviewer still has concern on the theoretical and empirical contribution of the work.

---

> > ### Comment · Reviewer_m2H3 · 2023-11-20
> >
> > I appreciate the refinement made to the conclusion of Theorem 1 by the authors. After a careful review of the proof, I find myself with additional concerns regarding its accuracy. Further intuition from the author on the proof would be beneficial, and I am willing to reconsider my rating if the proof holds true.
> >
> > I note the term $\frac{R}{n^2}$, presumably derived from concentration results in mean estimation where $||\hat{\mu} - \mu||_2\leq O(R/n^2)$. However, the convergence rate for mean estimation is typically $1/n$ instead of $1/n^2$, as also indicated in Theorem 3.3 in Huang et al., 2021, which the author refers to. Could the author clarify the rationale for the term being $\frac{R}{n^2}$ instead of $\frac{R}{n}$?

---

> > > ### Author Response · Authors · 2023-11-20
> > >
> > > Thanks for asking this clarifying question. Indeed, the term $diam(D) + R/n^2$ comes from the private mean estimation guarantee. The reviewer is right that the convergence rate for mean estimation is $1/n$ as per Theorem 3.3 in Huang et al., 2021. However, the constant is $diam(D)$ instead of $R$. We would like to clarify the notations in the two papers. For a dataset $D = \{x_1, \ldots, x_n\}$, we use $R$ to denote its maximum Euclidean norm, defined as $\max_{x\in D} ||x||$. This term corresponds to $r(D)$ in Huang et al 2021. We use $diam(D)$ to denote the diameter of $D$, defined as $\max_{x \in D} || x - \mu(D)||$. This term corresponds to $\omega(D)$ in Huang et al 2021.
> > >
> > > **Dependence on $diam(D)$**: Note that Theorem 3.3 of Huang et al 2021 shows that with high probability, the private estimation error scales as $\sqrt{d} \omega(D)/(n \varepsilon)$, which corresponds to $\sqrt{d} \cdot diam(D)/(n \varepsilon)$ instead of $\sqrt{d} R/(n \varepsilon)$.  In Theorem 1 (or more precisely Lemma 3) of our paper, we used a weaker version of their result, namely, we only use the fact that when $n \ge \sqrt{d}/\varepsilon$, the mean estimation error is at most $diam(D)$ with high probability. Hence our result is consistent with Huang et al 2021 in terms of the dependence on $diam(D)$.
> > >
> > > **Dependence on $R$**: The dependence on $r(D)$ (or $R$ in our notation) is not stated explicitly in the guarantee term in Theorem 3.3 of Huang et al 2021. Instead, it is hidden in logarithmic terms in the requirement on $n$ in the Theorem statement. In Theorem 3.3, the term $u$ is similar to $r(D)$ as it serves as an upper bound on the maximum norm of the data points. However, in our paper, to obtain an in-expectation bound on the estimation error instead of the high-probability bound in Huang et al 2021, we need to choose an appropriate failure probability $\beta$ and add an extra term $\beta R$ to the high probability bound since if the high probability bound fails, the error could be as large as $R$, hence contributing $\beta R$ to the overall expectation. We choose $\beta = 1/n^2$ in our paper and this results in the $R/n^2$ term in our estimation guarantee in Lemma 3, which carries over to the statement of Theorem 1 (see detailed in Section B.1). We note here that technically, we can choose $\beta$ to be any inverse polynomial of $n$ and further reduce the dependence on $R$. We choose $\beta = 1/n^2$ here mainly for ease of presentation.
> > >
> > > We added a discussion after Lemma 3 (mean estimation result) in our paper to clarify the above result.
> > >
> > > We hope this clarifies the reviewer’s concern on the accuracy of our result. We would be happy to answer any further questions from the reviewer.

---

> > > > ### Comment · Reviewer_m2H3 · 2023-11-21
> > > >
> > > > Thank you to the authors for addressing my concerns. I am pleased to amend my rating to a 6.

---

### Official Review · Reviewer_FnQ7 · 2023-10-31

**Soundness:** 3 good
**Presentation:** 3 good
**Contribution:** 3 good
**Rating:** 8
**Confidence:** 2

**Summary:**

This paper looks at the effect of feature pre-processing on the DP-SGD.
Specifically, they focus on the case of linear models and aim to show theoretically and empirically the impact of feature pre-processing.
First, they give a counter-example to show the error of DP-SGD must be proportional to the maximum norm of the features.
Then, they show that carefully pre-processing the features by subtracting the mean and shifting the bias by a private quantile estimation can reduce the error to be proportional to the dimension of the dataset.
The experiments show a clear improvement in accuracy while spending a minimal privacy budget on the feature pre-processing.

**Strengths:**

- The paper gives strong theoretical and empirical support for feature processing, a topic previously only reasoned about empirically.
- The privacy utility trade-off is a significant roadblock in the adoption of DP-SGD, and this paper gives a simple solution to improve this trade-off and give a nice increase in accuracy.
- The paper is very well written.

**Weaknesses:**

## Theory does not match experiments
The algorithm is significantly modified for the experiment section. It was clear that the algorithm analyzed was designed to make the theory work, and components were dropped or simplified for the experiments.

The part I am concerned about is the normalization of the features before computing the mean. The reason is that the normalization must be carried out in a private manner to allow for bounded sensitivity. The current version seems to assume a normalized dataset without accounting for any privacy budget to carry out the normalization.

### Note
I was unable to verify the proofs in this paper due to my lack of theoretical background. I apologize that I leave this verification to my fellow reviewers.

**Questions:**

Can the authors comment on the normalization issue?

---

> ### Author Response · Authors · 2023-11-19
>
> We thank the reviewer for the very positive feedback and acknowledging the contributions of our work. Below we address the concern the reviewer raised about the experiment part of our paper.
>
> ***[The algorithm is significantly modified for the experiment section. …… The part I am concerned about is the normalization of the features before computing the mean.]***
>
> Thanks for raising the valid concern. Both algorithms we use for the theoretical analysis and experimental evaluations follow the same procedure of (1) private mean estimation and translation of the features; (2) perform DPSGD on the new features. The main difference is on the private mean estimation step. In the modified algorithm (Algorithm 3), we normalize the features to a fixed norm and use the private Gaussian mechanism for computing the mean.
>
> We note that the modified algorithm with a fixed normalization norm satisfies differential privacy guarantee as claimed in Lemma 9. In experiments, we treat the normalization norm as a hyperparameter and search over a small set of values: {1, 10, 100, 1000}. We then report the maximum accuracy achieved within the set. We agree that this hyperparameter search might leak extra information about the dataset and this is not taken into account in our reported privacy parameter. However, ignoring the privacy leakage in the hyperparameter search step is commonly used in current empirical evaluations of private training algorithms, e.g., in (De et al 2022). In fact, how to best evaluate the privacy leakage for hyperparameter search is an active area of research, e.g., in (Papernot and Steinke 2022), which is beyond the scope of this work.
>
> Hence, while modifications to the algorithm are made in the experiment section, we believe the result still demonstrates the importance of feature pre-processing for private linear classification tasks, which is the main message of our work.
>
> Nicolas Papernot and Thomas Steinke *Hyperparameter Tuning with Renyi Differential Privacy* ICLR 2022

---

> > ### Comment · Reviewer_FnQ7 · 2023-11-22
> >
> > Thanks for the clarification. I think the fact that normalization is treated as a hyper parameter should be mentioned in the paper. It does feel like cheating, but I agree, other works also cheat in the same way.

---

### Official Review · Reviewer_eA3j · 2023-11-02

**Soundness:** 3 good
**Presentation:** 3 good
**Contribution:** 2 fair
**Rating:** 6
**Confidence:** 3

**Summary:**

This paper proposes a differentially private SGD algorithm which pre-processes the features before gradient descent steps. It is shown that, in linear classification problems and under certain regimes of privacy parameters, the new algorithm converges to a solution with lower optimality gap that the standard mini-batch DP-SGD algorithm without feature pre-processing. The theoretical improvement is supported by numerical experiments on image data sets.

**Strengths:**

* The paper's contribution is made very easy to understand by the side-by-side comparison between standard DP-SGD's optimality gap and the new algorithm's optimality gap. The paper is overall well-written.

* The example on why DP-SGD's excess loss must scale with $R$ is simple yet effective.

* There is an extensive set of numerical experiments supporting the theoretical results.

**Weaknesses:**

* On the improvement of Theorem 1 over Lemma 1, it appears that the comparison boils down to $\text{diam}(D)\sqrt{\text{rank}(M)\log(1/\delta)} + R\log n$ versus $R\sqrt{\text{rank}(M)\log(1/\delta)}$. While it is clear that, when the diameter is significantly smaller than $R$, the former's first term $\text{diam}(D)\sqrt{\text{rank}(M)\log(1/\delta)}$ is dominated by $R\sqrt{\text{rank}(M)\log(1/\delta)}$, there is a lack of discussion on whether/why $R\log n$ is also dominated by $R\sqrt{\text{rank}(M)\log(1/\delta)}$. The lower bound result, Theorem 2, did not address Theorem 1's second term scaling with $R\log n$ either.

* It is ambiguous whether the improvement afforded by feature pre-processing is specific to linear classification. The term "linear optimization" in the title and the sentence " *even* for the simple case of linear classification ..." in the abstract seem to suggest that there are reasons to believe the improvement can generalize to other settings. However, there is barely discussion of other settings in this paper. If there are reasons why feature pre-processing also improves over DP-SGD in other problems (or linear optimization in general), mentioning these reasons may significantly improve the paper's contribution.

* A few minor points on writing:
    * the phrase "maximum norm of features" in the abstract could be understood as "$\ell_\infty$ norm of features". It may help to say "maximum Euclidean norm" instead.
    * The letter $G$ in Lemma 1 is quite far from the previous mention of $G$. It might help to define $G$ as the Lipschitz constant again in the theorem statement.

**Questions:**

Corresponding to the two points under "Weaknesses":

* Does the claim that Theorem 1 improves over Lemma 1 require any condition on $\text{rank}(M)$, $\delta$, $n$, or other parameters?
* What is the relationship, if any, between the benefit of feature pre-processing in linear classification and the "importance of feature processing for differentially private linear optimization" as the title says?

---

> ### Author Response · Authors · 2023-11-19
>
> We thank the reviewer for the positive feedback and constructive comments. Below we discuss the technical questions raised by the reviewer.
>
> ***[Comparison between Lemma 1 (prior result) and Theorem 1 in different regimes of $rank(M), n, \delta$, and other parameters?]***
>
> Thanks for raising the important point. After a careful examination of our result, we are able to obtain an improved performance guarantee of our proposed algorithm DPSGD-F with a more involved analysis. We have marked the updated result (Theorem 1) and analysis (Appendix B.5) in red in the updated draft. In short, we are able to improve the dependence on $R$ and $diam(D)$ to roughly $(diam(D) + R/n^2) \cdot \sqrt{rank(M)}/(n\varepsilon)$, ignoring other constants that don’t depend on $R$ and $diam(D)$. In this form, our result strictly improves upon the prior result. When $diam(D) > R/n^2$, $diam(D)$ is the leading term in the optimality gap and the dependence on $R$ is at most logarithmic (only appears in the requirement on the $n$). When $diam(D) < R/n^2$, the dependence on $R$ is improved by a factor of $n^2$, which also improves upon the prior result. Moreover, the $\frac{R}{n^2}$ term can be reduced to any inverse polynomial function of $n$ by increasing the requirement on $n$ in the assumption by a constant factor. We use $R/n^2$ mainly for presentation purposes.
>
> ***[Relation between feature pre-processing for DP linear optimization and feature pre-processing for DP linear classification.]***
>
> This is a great question. We agree that the main focus of this paper is on linear classification models. However, since linear classification is a special case of linear optimization with specific requirements on the type of loss functions used, we believe our result also hints on the role of feature pre-processing for linear optimization problems. The main property that we use about linear classification tasks is that the final minimizer satisfies assumption 2, which requires that the linear model should predict values with different signs for at least one pair of the samples at the minimizer. This may not hold for other linear optimization problems in general, e.g., linear regression. We suspect that some processing of the labels might help linear regression to satisfy this condition, e.g., subtracting the mean from the labels, but this might need further investigation since the mean needs to be noised to guarantee privacy. We leave studying whether linear optimization can help for general linear optimization problems as important future directions.
>
> ***[Suggestions on improving the presentation.]***
>
> Thanks for the suggestions on improving the presentation. We have added an explanation for the norms used and revisited the definition of $G$ when necessary. Changes are marked in red.

---

### Meta-Review · Area_Chair_arLF · 2023-12-11

**Metareview:**

(a) Summarize the scientific claims and findings of the paper based on your own reading and characterizations from the reviewers.

For linear classification private feature preprocessing can help under differentially privacy. Theoretically, it is shown that there exists an example where, without feature preprocessing, DPSGD incurs a privacy error proportional to the maximum norm of features over all samples. DPSGD-F combines DPSGD with feature preprocessing and prove that for classification tasks, it incurs a privacy error proportional to the diameter of the features. In particular, it is shown that, in linear classification problems and under certain regimes of privacy parameters, the new algorithm converges to a solution with lower optimality gap that the standard mini-batch DP-SGD algorithm without feature pre-processing. The practicality of the proposed algorithm is demonstrated on image classification benchmarks.

(b) What are the strengths of the paper?

The paper is clear and well motivated. Experiments are extensive and support the theoretical claims. The toy example showing that DPSGD performance depends on R is effective.

(c) What are the weaknesses of the paper? What might be missing in the submission?

The privacy accounting for the preprocessing step is not properly done. Theorem 1 is still dimension dependent. The toy example is not convincing enough.

**Justification For Why Not Higher Score:**

The main idea is related to adaptive clipping, which has been explored in several other papers. The main novelty seems to be the fact that the authors analyze GLM that demonstrates the gain in adaptive clipping.

**Justification For Why Not Lower Score:**

The paper has made interesting connections between the geometry of the data, clipping, and GLMs. Shown in this particular way, the results are new and matches intuitions from experiments.

---

### Decision · Program_Chairs · 2024-01-16

Accept (poster)